# Towards Uncertainties in Deep Learning that Are Accurate and Calibrated

## Abstract

Predictive uncertainties can be characterized by two properties—calibration and sharpness. This paper introduces algorithms that ensure the calibration of any model while maintaining sharpness. They apply in both classification and regression and guarantee the strong property of distribution calibration, while being simpler and more broadly applicable than previous methods (especially in the context of neural networks, which are often miscalibrated). Importantly, these algorithms achieve long-standing statistical principle that forecasts should maximize sharpness subject to being fully calibrated. Using our algorithms, machine learning models can under some assumptions be calibrated without sacrificing accuracy: in a sense, calibration can be a free lunch. Empirically, we find that our methods improve predictive uncertainties on several tasks with minimal computational and implementation overhead.

## 1 Introduction

Probabilistic forecasts can be characterized by two properties—calibration and sharpness (Gneiting et al., 2007). Intuitively, calibration means that a 90% confidence interval contains the true outcome 90% of the time. Sharpness means that these confidence intervals are narrow. These qualities are grounded in the statistics literature on proper scoring rules, and are widely used to evaluate forecasts (Gneiting and Raftery, 2005; 2007) in domains such as medicine (Saria, 2018), robotic control (Malik et al., 2019), and human-in-the-loop learning (Werling et al., 2015).

This paper introduces algorithms that ensure the calibration of any predictive machine learning model while maintaining sharpness. They apply to both classification and regression tasks and guarantee the strong property of distribution calibration (which generalizes standard quantile and classification-based calibration; Song et al. (2019)) in any model, including deep learning models, which are often miscalibrated (Guo et al., 2017). Unlike existing methods for distribution calibration (Song et al., 2019), ours can be used with any model (not just ones that output Gaussians), are very simple to implement in differentiable programming frameworks, and have theoretical guarantees.

Importantly, our algorithms achieve for the first time a long-standing statistical principle that forecasts should "maximize sharpness subject to being calibrated" proposed by Gneiting et al. (2007). We prove that under some assumptions this principle is achievable in modern machine learning models in a black-box manner and without sacrificing overall performance. This lends strong support for this principle as a way of reasoning about uncertainty in machine learning.

In a sense, calibration is a rare free lunch in machine learning, and we argue that it should be enforced in predictive models and taken advantage of in downstream applications. Empirically, we find that our method consistently outputs well-calibrated predictions across a wide range of experiments, while improving performance on downstream tasks with minimal implementation overhead.

**Contributions.** In summary, we make three contributions. We propose a new recalibration technique that (a) is among the only to guarantee distribution calibration besides Song et al. (2019). Unlike Song et al. (2019) we can (b) recalibrate any parametric distribution (not just Gaussians) and (c) our method is simpler. While theirs is based on variational inference in Gaussian processes, ours uses a neural network that can be implemented in a few lines of code, which encourages adoption. Our method (d) applies to both classification and regression and (e) outperforms methods by Song et al. (2019) and Kuleshov et al. (2018) as well as Platt and temperature scaling.

We also formally prove that our technique produces asymptotically distributionally calibrated forecasts while minimizing regret. Most methods (e.g., Platt, Kuleshov scaling, Song et al., etc.) do not have a correctness proof, except for conformal prediction, which is significantly more complex.

Finally, our analysis formalizes the well-known paradigm of Gneiting et al. (2007) and provides the first method that provably achieves it. This lends strong support for this principle and influences how one should reason about uncertainty in machine learning. An important takeaway is that calibration can be achieved in most applications of machine learning with very little cost. As such, calibration can be a rare free lunch that we believe should be leveraged throughout machine learning.

## 2 BACKGROUND

### 2.1 PREDICTIVE UNCERTAINTY IN MACHINE LEARNING

Supervised machine learning models commonly predict a probability distribution over the output variables — e.g. class membership probabilities or the parameters of an exponential family distribution. These predictive uncertainties are useful for interpretability, safety, and downstream decision-making. Aleatoric uncertainty captures the inherent noise in the data, while epistemic uncertainty arises from not having a large enough dataset to estimate model parameters (Kendall and Gal, 2017).

**Notation.** Formally, we say that a machine learning forecaster $H : \mathcal{X} \to \Delta(\mathcal{Y})$ outputs a probability distribution $F(y) : \mathcal{Y} \to [0, 1]$ in the space $\Delta(\mathcal{Y})$ of distributions over $y$. We use $f$ to denote the probability density or probability mass function associated with $F$. The model $H$ is trained on a labeled dataset $x_t, y_t \in \mathcal{X} \times \mathcal{Y}$ for $t = 1, 2, ..., T$ of i.i.d. realizations of random variables $X, Y \sim \mathbb{P}$, where $\mathbb{P}$ is the data distribution.

### 2.2 WHAT DEFINES GOOD PREDICTIVE UNCERTAINTIES?

The standard tool in statistics for evaluating the quality of predictive uncertainties is a proper scoring rule (Gneiting and Raftery, 2007). Formally, a scoring rule $S : \Delta(\mathcal{Y}) \times \mathcal{Y} \to \mathbb{R}$ assigns a "score" to a probabilistic forecast $F \in \Delta(\mathcal{Y})$ and a realized outcome $y \in \mathcal{Y}$. Given a true distribution $G \in \Delta(\mathcal{Y})$ for $y$, we use the notation $S(F, G)$ for the expected score $S(F, G) = \mathbb{E}_{y \sim G} S(F, y)$.

We say that a score $S$ is proper if it is minimized by $G$ when $G$ is the true distribution for $y$: $S(F, G) \geq S(G, G)$ for all $F$. When $S$ is proper, we also refer to it as a proper loss. An example of a proper loss is the log-likelihood $S(F, y) = \log f(y)$, where $f$ is the probability density or probability mass function of $F$. Another common loss is the check score $\rho_\tau(y, f) = \tau(y - f)$ if $y \geq f$ and $(1 - \tau)(f - y)$ otherwise; it can be used to estimate the $\tau$-th quantile of a distribution. See Table 1 for additional examples.

What are the qualities of a good probabilistic prediction, as measured by a proper scoring rule? It can be shown that every proper score is a sum of the following terms (Gneiting et al., 2007):

$$\text{proper loss} = \text{calibration} \underbrace{-\text{sharpness} + \text{irreducible term}}_{\text{refinement term}}.$$

Thus, there are precisely two qualities that define an ideal forecast: calibration and sharpness. We examine each of them next.

### 2.3 CALIBRATION AND SHARPNESS — TWO QUALITIES OF AN IDEAL PREDICTION

Formally, calibration can be defined by the equation

$$\mathbb{P}(Y = y \mid F_X = F) = f(y) \text{ for all } y \in \mathcal{Y}, F \in \Delta(\mathcal{Y}), \tag{1}$$

where $X, Y \sim \mathbb{P}$ are random variables corresponding to the input features and targets, and $F_X = H(X)$ is the forecast at $X$, itself a random variable that takes values in $\Delta(\mathcal{Y})$. We use $f$ to denote the probability density or probability mass function associated with $F$.

When $\mathcal{Y} = \{0, 1\}$ and $F_X$ is a Bernoulli distribution with parameter $p$, we can write (1) as $\mathbb{P}(Y = 1 \mid F_X = p) = p$. This has a simple intuition: the true probability of $Y = 1$ is $p$ conditioned

| Proper Score | Loss $L(F, G)$ | Calibration $L_c(F, Q)$ | Refinement $L_r(Q)$ |
|---|---|---|---|
| Logarithmic | $\mathbb{E}_{y \sim G} \log f(y)$ | $\text{KL}(q \| f)$ | $H(q)$ |
| CRPS | $\mathbb{E}_{y \sim G} (F(y) - G(y))^2$ | $\int_{-\infty}^{\infty} (F(y) - Q(y))^2 \mathrm{d}y$ | $\int_{-\infty}^{\infty} Q(y)(1 - Q(y)) dy$ |
| Quantile | $\mathbb{E}_{y \sim G}^{\tau \in U[0,1]} \rho_\tau(y - F^{-1}(\tau))$ | $\int_0^1 \int_{Q^{-1}(\tau)}^{F^{-1}(\tau)} (Q(y) - \tau) dy d\tau$ | $\mathbb{E}_{y \sim Q}^{\tau \in U[0,1]} \rho_\tau(y - Q^{-1}(\tau))$ |

Table 1: Proper loss functions. A proper loss is a function $L(F, G)$ over a forecast $F$ targeting a variable $y \in \mathcal{Y}$ whose true distribution is $G$ and for which $S(F, G) \geq S(G, G)$ for all $F$. Each $L(F, G)$ decomposes into the sum of a calibration loss term $L_c(F, Q)$ (also known as reliability) and a refinement loss term $L_r(Q)$ (which itself decomposes into a sharpness and an uncertainty term). Here, $Q(y)$ denotes the cumulative distribution function of the conditional distribution $\mathbb{P}(Y = y \mid F_X = F)$ of $Y$ given a forecast $F$, and $q(y), f(y)$ are the probability density functions of $Q$ and $F$, respectively. We give three examples of proper losses: the log-loss, the continuous ranked probability score (CRPS), and the quantile loss.

on predicting it as $p$. Equation 1 extends beyond binary classification to arbitrary distributions. For example, if $F$ is a Gaussian with variance $\sigma^2$, this definition asks that the data distribution conditioned on predicting $F$ also has variance $\sigma^2$. This recently proposed definition is called *distribution* calibration (Song et al., 2019).

A closely related, but weaker concept is quantile calibration (Kuleshov et al., 2018), which asks that a 90% confidence interval contains the true value 90% of the time. Formally, it can be written as: $\mathbb{P}(Y \leq \text{CDF}_{F_X}^{-1}(p)) = p$ for all $p \in [0, 1]$, Quantile calibration is implied by distributional calibration (Song et al., 2019).

Calibration by itself is not sufficient to produce a useful forecast. For example, it is easy to see that a binary classifier that always outputs $\mathbb{P}(Y = 1)$ as the probability that $Y = 1$ is calibrated; however it does not even use the features $X$ and thus cannot be accurate.

In order to be useful, forecasts must also be *sharp*. Intuitively, this means that predicted confidence intervals should be as tight as possible around a single value. This is captured by proper scoring rules as part of a refinement term (see Table 1), which equals an irreducible term minus a sharpness term (Murphy, 1973; Brocker, 2009). The latter is maximized when we minimize the scoring rule.

**Are Modern Machine Learning Models Calibrated And Sharp?** Most machine learning models are not calibrated out-of-the-box (Niculescu-Mizil and Caruana, 2005; Guo et al., 2017). Two reason for this are the limited expressivity of the model $H$—we cannot perfectly fit the entirety of the level curves of the data distribution—and computational approximations—computing extract predictive uncertainties may be intractable, and approximations are not entirely accurate.

A final reason stems from how models are trained—since we cannot fit a perfect $H$, standard objective functions induce a tradeoff between sharp and calibrated forecasts. Next, we will show that by training models differently, we can achieve calibration without sacrificing performance.

## 3 ENSURING DISTRIBUTION CALIBRATION IN ANY MODEL

This section introduces algorithms that ensure the distirbutional calibration of any predictive machine learning model while maintaining sharpness. Unlike existing methods for distribution calibration, ours can be used with any model (not just ones that output Gaussians), are very simple to implement in differentiable programming frameworks, and have theoretical guarantees.

We first assume there exists a parameterization $\Phi_1$ of the probabilities returned by forecaster $H$: for each $p \in \Delta(\mathcal{Y})$ returned by $H$, there exist parameters $\phi \in \Phi_1$ that describe $p$. The $\phi$ can be the natural parameters of an exponential family distribution, such as $(\mu, \sigma^2)$ describing a Gaussian.

We consider a class of algorithms based on a classic approach called recalibration. First, we train a base forecaster $H$ to minimize a proper loss $L$. Then, we train an auxiliary model $R : \Phi_1 \rightarrow \Phi_2$ (called the recalibrator) over the outputs of $H$ that outputs the parameters $\phi_2 \in \Phi_2$ of another

distribution such that $L$ is minimized. Here $\Phi_2$ is a second parameterization of $\Delta(\mathcal{Y})$ (possibly the same). As a result, the forecasts $(R \circ H)(X)$ will be calibrated. We provide details in Algorithm 1.

---

**Algorithm 1** Calibrated Learning of Probabilistic Models.

---

**Input:** Model $H : \mathcal{X} \to \Phi_1$, recalibrator $R : \Phi_1 \to \Phi_2$, training set $\mathcal{D}$, recalibration set $\mathcal{C}$
**Output:** Recalibrated model $R \circ H : \mathcal{X} \to \Phi_2$.

1. Fit the base model on $\mathcal{D}$: $\min_H \sum_{(x,y) \in \mathcal{D}} L(H(x), y)$

2. Fit the recalibration model $R$ on the output of $H$ on $\mathcal{C}$: $\min_R \sum_{(x,y) \in \mathcal{C}} L\left((R \circ H)(x), y\right)$

---

Implementing this approach requires choosing parameterizations of probabilities $\Phi_1, \Phi_2$, a recalibration model $R$, and an objective $L$. We discuss these choices below; then we clarify how 1 differs from existing recalibration algorithms.

**Parameterizing Probability Distributions.** When fitting a classification model, each distribution $\Delta(\mathcal{Y})$ is a categorical and can be parameterized via its $K \geq 2$ class membership probabilities. In regression, most widely used models such as neural networks already output parameterized probabilities, in which the $\phi$ are usually the natural parameters of an exponential family model.

In the most general case, if we only have black-box access to a density function or a cumulative distribution function, we may form a $d$-dimensional representation by evaluating the distribution at a grid of $d$ points. For example, if we have black-box access to a quantile function $F^{-1}$, we may featurize $F$ via its sequence of quantiles $\phi(F) = (F^{-1}(\alpha_i))_{i=1}^d$ for some sequence of $d$ levels $\alpha_i$, possibly chosen uniformly in $[0, 1]$.

In addition to the above techniques, the representation of output probabilities in $\Phi_2$ coming from $R$ can leverage flexible invertible models of the CDF, following methods developed in the normalizing flow literature, including monotonic neural networks, sum-of-squares polynomials (Wehenkel and Louppe, 2019; Jaini et al., 2019) spline functions (Muller et al., 2019; Durkan et al., 2019), piecewise separable models (Wehenkel and Louppe, 2019), and others.

**Choosing a Recalibrator.** Ideal recalibrators are highly effective at optimizing the proper loss $L$ (see Section 4). In a simple setting like binary classification, our task reduces to one-dimensional density estimation; in such cases we can provably achieve calibration asymptotically by using kernel density estimation for the recalibrator $R$, while controlling the kernel width as a function of the dataset size to trade off overfitting and underfitting (Wasserman, 2006). In regression settings, we may rely on other non-parametric techniques such as Gaussian processes.

An alternative approach is to rely on expressive neural networks; although their optimization is a non-convex, they are very effective at fitting proper losses $L$, feature mature regularization techniques, and can be implemented easily within deep learning frameworks, possibly within the same computation graph as a neural forecaster $H$, which can simplify deployment.

In the classification setting, a natural architecture for $R$ is a sequence of dense layers mapping the simplex $\Delta_K$ into $\Delta_K$. In regression settings, $R$ needs to output a density function: a natural architecture for this is a mixture density network (MDN; Bishop (1994)).

**Choosing a Proper Loss** A natural choice of proper loss is the log-loss. It applies in both calibration and regression; optimizing it is a standard supervised learning problem.

In regression settings, we found that using the quantile loss $L = \mathbb{E}_{\tau \in U[0,1]} \mathbb{E}_{y \sim G} \rho_\tau(y - F^{-1}(\tau))$ (see Table 1) was numerically stable and produced the best performance. This objective fits a model $R_\theta(\tau; \phi)$ to estimate the $\tau$-th conditional quantile $F^{-1}(\tau)$ at $\phi$. When $R_\theta(\tau; \phi)$ is a neural network that takes in $\tau$ and $x$, we minimize the quantile loss $\mathbb{E}_{\tau \in U[0,1]} \mathbb{E}_{y \sim G} \rho_\tau(y - F^{-1}(\tau))$ using gradient descent, approximating both expectations using Monte Carlo (details are in the experiments section).

**Comparing Against Song et al.** Interestingly, the method of Song et al. (2019) is a special case of ours when $\Phi_1$ consists of Gaussian natural parameters, $\Phi_2$ consists of parameters for the Beta link function, $R$ is a Gaussian process, and $L$ is the log-likelihood. However, the resulting problem can

only be solved using variational inference, which is slow and complex to implement. Our framework instead admits simple solutions based on gradient descent.

**Comparing Against Other Approaches.** Algorithm 1 performs recalibration like many previous methods (e.g., Platt, Kuleshov et al., Song et al., etc.); it may thus appear to be the same as these methods. However, that is not the case. First, existing recalibration approaches operate over the space of probabilities (class probabilities or CDF values); ours operates over *functional parameters*. This is what enables it to achieve distribution rather than quantile calibration (Kuleshov et al., 2018).

Our approach also involves novel recalibration objectives (e.g., the quantile divergence in regression) which differ from the calibration error of Kuleshov et al. (2018). We also use different types of models (small neural networks instead of isotonic regression) and different optimization procedures used (stochastic gradient descent instead of variational inference). Thus, our recalibration strategy is distinct from previous work.

## 4 THEORETICAL ANALYSIS

Next, we now show that under some assumptions calibration is provably achievable in modern machine learning models in a black-box manner and without sacrificing overall performance. In that sense, calibration is a rare *free lunch* in machine learning.

We start with some notation. We have a recalibration dataset of size $T$ sampled from $\mathbb{P}$ and train a recalibrator $R : \Delta(\mathcal{Y}) \to \Delta(\mathcal{Y})$ over the outputs of a base model $H$ to minimize a proper loss $L$. We denote the Bayes-optimal recalibrator by $B := \mathbb{P}(Y = y \mid H(X))$; the distribution of $Y$ conditioned on the forecast $(R \circ H)(X)$ is $Q := \mathbb{P}(Y = y \mid (R \circ H)(X))$. We are interested in expectations of various losses over $X, Y$; to simplify notation, we omit the variable $X$, e.g. $\mathbb{E}[L(R \circ H, Y)] = \mathbb{E}[L(R(H(X)), Y)]$.

Next, we will assume the following condition under which Algorithm 1 works.

**Assumption 1.** The model $R$ can minimize expected risk such that w.h.p. we have

$$\mathbb{E}[L(B \circ H, Y)] \leq \mathbb{E}[L(R \circ H, Y)] < \mathbb{E}[L(B \circ H, Y)] + \delta$$

where $\delta > 0$, $\delta = o(T)$ is a bound that decreases with $T$ and $\mathbb{E}[L(B \circ H, Y)]$ is the irreducible loss.

This assumption implies that the recalibrator can perform density estimation in what is usually a small number of dimensions (one or two). For some recalibrators, e.g., neural nets, it may not provably hold (e.g., because of non-convexity). However, neural networks are effective density estimators in practice, and we can quantify whether they estimate density well on a hold-out set. This assumption provably holds for many non-parametric density estimation methods.

**Fact 1** (Wasserman (2006)). When $R$ implements kernel density estimation and $L$ is the log-loss, Assumption 1 holds with $\delta = o(1/T^{2/3})$.

We now prove two key lemmas. We show that Algorithm 1 outputs calibrated forecasts without reducing the performance of the base model, as measured by regret relative to loss $L$.

**Lemma 1.** *The model $R \circ H$ is asymptotically calibrated, in the sense that $\mathbb{E}[L_c(R \circ H, Q)] < \delta$ for $\delta = o(T)$ w.h.p.*

*Proof.* Recall that the loss $\mathbb{E}[L(R \circ H, Y)]$ decomposes into a sum of calibration and refinement terms $\mathbb{E}[L_c(R \circ H, Q)] + \mathbb{E}[L_r(Q)]$ where $Q(y) := \mathbb{P}(Y = y \mid (R \circ H)(X))$.

As shown by Kull and Flach (2015), refinement further decomposes into a group loss and an irreducible term: $\mathbb{E}[L_r(Q)] = \mathbb{E}[L_g(Q, B \circ H)] + \mathbb{E}[L(B \circ H, Y)]$, where $B(Y = y \mid H(X))$ is the Bayes-optimal recalibrator. The form of the group loss $L_g$ is the same as that of $L_c$. We may then write:

$$\underbrace{\mathbb{E}[L(B \circ H, Y)]}_{\text{Bayes-Optimal Loss}} \leq \underbrace{\mathbb{E}[L_c(R \circ H, Q)]}_{\text{Calibration Loss}} + \underbrace{\mathbb{E}[L_g(Q, B \circ H)]}_{\text{Group Loss}} + \underbrace{\mathbb{E}[L(B \circ H, Y)]}_{\text{Bayes-Opt Loss}}$$

$$= \underbrace{\mathbb{E}[L(R \circ H, Y)]}_{\text{Proper Loss}} < \underbrace{\mathbb{E}[L(B \circ H, Y)]}_{\text{Bayes-Optimal Loss}} + \delta$$

where $\delta > 0, \delta = o(T)$. In the first equality we used the decomposition of Kull and Flach (2015) and in the last inequality we used Assumption 1. It follows that $\mathbb{E}[L_c(R \circ H, Q)] < \delta$, i.e. the calibration loss is small. $\square$

**Lemma 2.** *The recalibrated model is asymptotically as good as the base model:* $\mathbb{E}[L(R \circ H, Y)] \leq \mathbb{E}[L(H, Y)] + \gamma$, *where* $\gamma > 0, \gamma = o(T)$ *is a bound that decreases with* $T$.

*Proof.* The claim holds by empirical risk minimization, since $R \circ H$ minimizes $L$, but is more expressive than $H$ and $R$ can represent the identity map (by Assumption 1). $\square$

We now combine these two lemmas to show Algorithm 1 ensures calibration and low regret.

**Theorem 3.** *Algorithm 1 produces a model that minimizes expected risk, while w.h.p. achieving asymptotically optimal calibration.*

*Proof.* The base model $H$ is trained using empirical risk minimization (ERM). The model $R \circ H$ minimizes the same objective $L$, hence minimizes the same expected risk by ERM theory. Also, by Lemma 2, the expected risk of $R \circ H$ also asymptotically approaches a lower value as that of $H$.

By Lemma 1, the model $R \circ H$ produces asymptotically calibrated forecasts w.h.p. $\square$

Thus, given enough data, we are guaranteed to produce calibrated forecasts and preserve base model performance (as measured by $L$). Thus, calibration is a property that can be achieved in most applications of machine learning with almost no cost. As such, calibration is a rare free lunch in machine learning.

**Finite-Sample Bounds.** Note that our analysis provides *finite-sample* and not only asymptotic bounds on the regret and calibration error—the bounds are stated in terms of variables $\delta$, and $\gamma$ that are each $o(T)$. The bound $\delta$ on the calibration error directly depends on the finite-sample bound on the generalization error of the algorithm used as the recalibrator.

**Practical Considerations.** Assumption 1 suggests that we want to use a model family that can minimize the expected risk $\mathbb{E}[L(H(X), Y)]$ well. Thus, in practice we want to select a highly flexible algorithms for which we can control overfitting and underfitting. This motivates our earlier advice of using density estimation algorithms—which have provable guarantees—and neural networks—which are expressive and feature effective regularization techniques

## 5 WHAT UNCERTAINTIES ARE NEEDED IN MODERN DEEP LEARNING?

Good predictive uncertainties are calibrated and sharp and these two properties yield optimal values of the log-likelihood and other proper loss functions. Thus, they characterize an ideal forecast. In practice, however, modern machine learning models do not output such ideal predictions. What then is the ideal type of forecast that we should aim to obtain from our models?

Gneiting et al. (2007) argue that predictive uncertainties should be maximally sharp subject to being calibrated. They propose a diagnostic approach based on this principle; this approach is commonly used in statistics for *evaluating* the predictive performance of probabilistic models.

In this paper, we also argue for this general principle, but approach it in a *prescriptive* way — we claim that this principle should be enforced in modern ML systems, and we show how to do so. Specifically, we show that any model can be modified to output calibrated uncertainties, and this property can be provably achieved without sacrificing performance.

### 5.1 CALIBRATED RISK MINIMIZATION

We formalize the intuition behind "maximizing sharpness subject to being calibrated" as follows. We argue that we should be training models to minimize expected risk (as measured by a proper loss) subject to being perfectly calibrated. We call this principle calibrated risk minimization.

**Definition 5.1** (Calibrated Risk Minimization)**.** We select a model $H$ that minimizes the constrained expected risk

$$\min_H \mathbb{E}[L(H(X), Y)] \text{ subject to } \mathbb{E}[L_c(H(X), Q))] = 0,$$

where $L$ is a proper scoring rule, $L_c$ is its associated calibration loss derived from the calibration-reliability decomposition of $L$, and $Q(y) := \mathbb{P}(Y = y \mid F_X = F)$.

Recall that a proper loss $L(F, G)$ decomposes into a sum of calibration $L_c(H(X), Q)$ and reliability $L_r(Q)$; the latter equals an irreducible term minus sharpness. Thus, by minimizing $L(F, G)$ subject to $L_c(H(X)) = 0$, we are maximizing sharpness subject to calibration (Gneiting et al., 2007).

Note that a special case of the above principle is *calibrated maximum likelihood*, in which we seek a model that maximizes the expected log-likelihood $\mathbb{E}_{X,Y} \log F_X(Y)$ under the calibration constraint that $\mathbb{E}_{X,Y} \mathrm{KL}(Q \mid\mid F) = 0$.

Machine learning models are normally trained to minimize expected risk; our principle asks that in addition they should be calibrated. Our main result (Theorem 3) shows that this criterion is achievable.

## 5.2 Why Do We Need Calibrated And Sharp Uncertainties?

Probabilistic models are important building blocks of machine learning systems in many domains—including medicine, robotics, industrial automation, and others. Calibration is not difficult to achieve in many of these domains; hence, we argue that it should be enforced in predictive models, which will unlock the following set of benefits in downstream applications.

**Safety and Interpretability.** Good predictive uncertainties are important for model interpretability: in user-facing applications, humans make decisions based on model outputs and need to assess the confidence of the model, for example when interpreting an automated medical diagnosis. Calibration is also important for model safety: in areas such as robotics, we want to minimize the probability of adverse outcomes (e.g., a crash), and estimating these outcomes' probabilities is an important step for that (Berkenkamp et al., 2017).

**Model-Based Planning.** More generally, good predictive uncertainties also improve downstream decision-making applications such as model-based planning (Malik et al., 2019), a setting in which agents learn a model of the world to plan future decisions (Deisenroth and Rasmussen, 2011). Planning with a probabilistic model improves performance and sample complexity, especially when representing the model using a deep neural network. and improves the cumulative reward and the sample complexity of model-based agents (Rajeswaran et al., 2016; Chua et al., 2018).

**Efficient Exploration.** Balancing exploration and exploitation is a common challenge in many applications on machine learning such as reinforcement learning, Bayesian optimization, and active learning. When probabilistic models are uncalibrated, inaccurate confidence intervals might incentivize the model to explore ineffective actions, degrading performance. Calibrated uncertainties have been shown to improve decision-making in bandits (Malik et al., 2019) and likely to extend to Bayesian optimization and active learning as well.

**Other Applications.** The importance of accurate confidence estimates has been highlighted by practitioners in many fields, including medicine (Saria, 2018), meteorology (Raftery et al., 2005), and natural language processing (Nguyen and O'Connor, 2015). Accurate confidence estimates also play an important in computer vision applications, such as depth estimation (Kendall and Gal, 2017).

## 6 Experiments

### 6.1 Setup

**Datasets.** We use a number of UCI regression datasets varying in size from 194 to 8192 training instances; each training input may have between 6 and 159 continuous features. We randomly use 25% of each dataset for testing, and use the rest for training. We also perform image classification on the following standard datasets: MNIST, SVHN, CIFAR10.

**Bayesian Linear Regression**

| dataset | Uncalibrated MAE | MAPE | CHK | Kuleshov et al. MAE | MAPE | CHK | Song et al. MAE | MAPE | CHK | Ours MAE | MAPE | CHK |
|---|---|---|---|---|---|---|---|---|---|---|---|---|
| mpg | 2.456 | 0.114 | 0.921 | 2.465 | 0.114 | 0.916 | 2.498 | 0.113 | 0.915 | 2.398 | 0.113 | 0.902 |
| boston | 3.459 | 0.171 | 1.392 | 3.399 | 0.165 | 1.365 | 3.387 | 0.164 | 1.372 | 3.349 | 0.169 | 1.312 |
| yacht | 6.171 | 3.800 | 2.438 | 5.964 | 4.802 | 2.379 | 1.502 | 1.326 | 1.271 | 0.908 | 0.336 | 0.352 |
| wine | 0.628 | 0.106 | 0.240 | 0.627 | 0.105 | 0.239 | 0.637 | 0.106 | 0.241 | 0.628 | 0.106 | 0.240 |
| crime | 0.516 | 0.087 | 0.202 | 0.514 | 0.086 | 0.202 | 0.526 | 0.092 | 0.205 | 0.516 | 0.086 | 0.198 |
| auto | 0.635 | 0.058 | 0.251 | 0.629 | 0.057 | 0.250 | 0.635 | 0.062 | 0.258 | 0.636 | 0.058 | 0.250 |
| cpu | 39.896 | 0.647 | 15.871 | 39.086 | 0.636 | 15.463 | 35.166 | 0.462 | 13.915 | 28.160 | 0.324 | 13.615 |
| bank | 39.508 | 0.590 | 17.478 | 39.148 | 0.568 | 16.639 | 34.115 | 0.424 | 15.426 | 29.314 | 0.387 | 14.274 |

**Bayesian Neural Network**

| dataset | Uncalibrated MAE | MAPE | CHK | Kuleshov et al. MAE | MAPE | CHK | Song et al. MAE | MAPE | CHK | Ours MAE | MAPE | CHK |
|---|---|---|---|---|---|---|---|---|---|---|---|---|
| mpg | 2.736 | 0.122 | 1.198 | 2.973 | 0.127 | 1.176 | 2.678 | 0.118 | 1.101 | 2.601 | 0.119 | 1.083 |
| boston | 2.966 | 0.147 | 1.237 | 3.003 | 0.141 | 1.206 | 3.305 | 0.171 | 1.181 | 2.983 | 0.144 | 1.404 |
| yacht | 3.539 | 0.592 | 1.535 | 3.772 | 0.516 | 1.519 | 3.375 | 0.498 | 1.511 | 3.175 | 0.470 | 1.510 |
| wine | 0.625 | 0.105 | 0.252 | 0.630 | 0.104 | 0.241 | 0.623 | 0.109 | 0.239 | 0.621 | 0.103 | 0.238 |
| crime | 0.498 | 0.085 | 0.195 | 0.487 | 0.083 | 0.192 | 0.488 | 0.082 | 0.192 | 0.491 | 0.083 | 0.192 |
| auto | 0.625 | 0.059 | 0.250 | 0.623 | 0.058 | 0.246 | 0.640 | 0.060 | 0.248 | 0.644 | 0.061 | 0.248 |
| cpu | 74.001 | 0.518 | 35.528 | 71.033 | 0.633 | 28.683 | 68.428 | 0.641 | 31.810 | 66.428 | 0.630 | 31.318 |
| bank | 96.088 | 0.722 | 46.022 | 90.887 | 1.105 | 44.101 | 87.096 | 1.103 | 41.192 | 85.096 | 0.822 | 39.257 |

**Deep Ensemble**

| dataset | Uncalibrated MAE | MAPE | CHK | Kuleshov et al. MAE | MAPE | CHK | Song et al. MAE | MAPE | CHK | Ours MAE | MAPE | CHK |
|---|---|---|---|---|---|---|---|---|---|---|---|---|
| mpg | 7.667 | 0.288 | 3.556 | 11.858 | 0.471 | 5.207 | 9.573 | 0.364 | 3.639 | 9.010 | 0.358 | 3.538 |
| boston | 8.427 | 0.328 | 3.820 | 12.860 | 0.508 | 5.495 | 8.132 | 0.321 | 3.712 | 8.004 | 0.318 | 3.623 |
| yacht | 9.406 | 0.774 | 4.604 | 10.836 | 1.684 | 4.932 | 9.459 | 2.211 | 4.312 | 9.359 | 2.012 | 4.278 |
| wine | 0.715 | 0.132 | 0.308 | 0.701 | 0.122 | 0.276 | 0.695 | 0.128 | 0.269 | 0.690 | 0.123 | 0.268 |
| crime | 0.685 | 0.115 | 0.255 | 0.779 | 0.124 | 0.344 | 0.684 | 0.113 | 0.251 | 0.686 | 0.115 | 0.252 |
| auto | 0.862 | 0.084 | 0.309 | 0.862 | 0.084 | 0.314 | 0.878 | 0.091 | 0.352 | 0.868 | 0.088 | 0.301 |
| cpu | 58.101 | 0.505 | 20.095 | 57.333 | 0.543 | 18.352 | 57.982 | 0.540 | 18.810 | 55.428 | 0.520 | 17.448 |
| bank | 59.088 | 0.512 | 21.216 | 58.731 | 0.509 | 21.087 | 57.493 | 0.498 | 20.596 | 55.869 | 0.482 | 19.196 |

Table 2: Calibration and accuracy on UCI regression datasets. We evaluate Bayesian linear regression, Bayesian neural networks, and deep ensembles using mean average error (MAE), mean absolute percent error (MAPE), and the check score (CHK); we compare against Kuleshov et al. (2018) and Song et al. (2019).

**Models.** Our first model is Bayesian Ridge Regression (MacKay, 1992). It uses a spherical Gaussian prior over the weights and a Gamma prior over the precision parameter. Posterior inference is performed in closed form as the prior is conjugate.

We also test a number of deep neural networks. We use variational dropout (Gal and Ghahramani, 2016) to produce probabilistic predictions. In our UCI experiments, we use fully-connected feedforward neural networks with two layers of 128 hidden units with a dropout rate of 0.5 and parametric ReLU non-linearities. We use convolutional neural networks (CNNs) on the image classification tasks. These are formed by fine-tuning a ResNet50 architecture on the training split for each dataset.

We also compare against a popular uncertainty estimation method recently developed specifically for deep learning models: deep ensembles (Lakshminarayanan et al., 2017). Deep ensembles average the predictive distributions of multiple models; we ensembled 5 neural networks, each having the same architecture as our standard model.

Our recalibrator $R$ was also a densely connected neural network with two fully connected hidden layers of 20 units each and parametric ReLU non-linearities. We added dense skip connections between the layers. In regression experiments, we featurized input distributions $F$ using nine quantiles $[0.1, ..., 0.9]$. We trained $R$ using the quantile regression version of Algorithm 1; we concatenated the quantile parameter $\tau \in [0, 1]$ to the featurization of $F$. In classification experiments, the inputs and the ouputs of $R$ are class probabilities, and $R$ is trained using the log-likelihood maximization version of Algorithm 1. All other architectural details are unchanged.

We did not observe significant overfitting in our experiments. We believe overfitting is mitigated by the fact that we perform quantile regression and thus learn a complex distribution function that is not easy to overfit.

## 6.2 REGRESSION EXPERIMENTS ON UCI DATA

We report the results of our regression experiments on the UCI datasets in Table 2. We evaluate the quality of forecasts using a check score $\rho_\tau(y, f) = \tau(y - f)$ if $y \geq f$ and $(1 - \tau)(f - y)$ as in Song et al. (2019); we average it over nine quantile levels $\tau \in 0.1, ..., 0.9$. We measure regression performance using the mean absolute percent error and mean average error.

Our method improves over the accuracies and uncertainties of Kuleshov et al. (2018), and in many cases over those of Song et al. (2019) on Bayesian linear regression, Bayesian neural networks, and deep ensembles, without ever being worse. Note that also that out method is simpler and easier to implement than that of Song et al. (2019) (it does not require implementing variational inference), and applies to any input distribution, not just Gaussians.

## 6.3 CLASSIFICATION EXPERIMENTS ON MNIST, SVHN, CIFAR10

We report the results of the image classification experiments in Table 3. We measure performance using accuracy and calibration error of Kuleshov et al. (2018) on the test set. We report these metrics for baseline and calibrated versions of convolutional neural network classifier. We perform recalibration with a simple softmax regression (a multi-class generalization of Platt scaling) and with the neural network recalibrator. The best uncertainties are produced by our method. Recalibrated and base models achieve similar levels of accuracy.

|  | MNIST | SVHN | CIFAR10 |
|---|---|---|---|
| **Base Model** | | | |
| Accuracy | 0.9952 | 0.9508 | 0.9179 |
| Calibration | 0.3166 | 0.5975 | 0.5848 |
| **Platt Scaling** | | | |
| Accuracy | 0.9952 | 0.9508 | 0.9181 |
| Calibration | 0.2212 | 0.3278 | 0.2233 |
| **Ours** | | | |
| Accuracy | 0.9951 | 0.9509 | 0.9163 |
| Calibration | 0.1030 | 0.2674 | 0.1091 |

Table 3: Performance on Image Classification

## 7 PREVIOUS WORK

**Probabilistic Forecasting.** More modern discussions of probabilistic forecasting can be found in the literature on meteorology (Gneiting and Raftery, 2005). This influential work appears in methods weather forecasting applications systems (Raftery et al., 2005). Most previous work focuses on classification, but recent work (Gneiting et al., 2007; Kuleshov et al., 2018) extends classical methods to regression.

Probabilistic forecasting has been studied extensively in the statistics literature (Murphy, 1973; Dawid, 1984), mainly in the context of evaluation using proper scoring rules (Gneiting and Raftery, 2007). Proper scores measure calibration and sharpness in classification (Murphy, 1973) and regression (Hersbach, 2000).

**Calibration.** Recalibration is a widely used approach for improving probabilistic forecasts. It takes it roots in the classification setting, where Platt scaling (Platt, 1999) and isotonic regression (Niculescu-Mizil and Caruana, 2005) are two widely used algorithms. The have been extended to multi-class (Zadrozny and Elkan, 2002), structured (Kuleshov and Liang, 2015), and online prediction Kuleshov and Ermon (2017). There is significant recent interest in calibration in deep learning (Guo et al., 2017; Lakshminarayanan et al., 2017; Gal et al., 2017; Kuleshov et al., 2018).

## 8 CONCLUSION

We take inspiration from the statistics literature and argue that predictive uncertainties should be evaluated by proper scoring rules, which measure two specific qualities of probabilistic predictions: calibration and sharpness. Gneiting et al. (2007) argued that predictive uncertainties should maximize calibration subject to sharpness and used this paradigm to evaluate forecasts. We formalize the paradigm of Gneiting et al. (2007) into a novel learning principle called calibrated risk minimization and propose a general algorithm that meets the requirements of this paradigm. Overall, we show that calibration is a property that can be achieved in predictive models with almost no cost. As such, calibration is a rare free lunch that should be enforced throughout applications of machine learning.

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
