# OpenReview forum: "Towards Uncertainties in Deep Learning that Are Accurate and Calibrated"
_ICLR.cc/2022/Conference — ICLR 2022 Submitted_

### Official Review · Reviewer_GeKX · 2021-11-01

**Correctness:** 2
**Technical Novelty And Significance:** 2
**Empirical Novelty And Significance:** 3
**Recommendation:** 3
**Confidence:** 3

**Main Review:**

Overall, I liked this paper. It is well-written in the sense that the text feels clear and readable, and the authors do a good job of situating their work with respect to calibration literature and recapitulating important results.

I have a few reservations and concerns which currently prevent me from recommending acceptance.

First, my worry is about what precisely we are calibrated with respect to. Imagine, for example, the algorithm involved taking D = {empty set}, H = identity, and R = {all training data}. At first glance, this ought to result in calibrated models, while it is also identical to standard practice. Can you explain to me what I'm missing here? Why does splitting the calibration into two non-trivial steps make the difference? How would you explain why standard training with a proper loss does not result in a calibrated model?

Second, there are other kinds of uncertainty that seem not to be represented in your model, which are what Bayesian deep learning tries to address.
For example, it seems straightforward to overfit the 'calibration' to C, and then be poorly calibrated on some other test data.
The algorithm might not be solving some sort of generalizable calibration problem, but quite a specific one.
In general, it would be useful to engage more on prior work about uncertainty in deep learning.

Third, there is a bit of a jump between your lemmas ant Theorem 3 which make claims entirely asymptotically and probabilistically and then your assertion that perfect calibration is available at almost no cost as a `free lunch'.
This is also reflected in your definition of CRM (5.1), which builds off the idea of having zero expected calibration loss, but we don't know how to actually enforce that constraint, we can only bound the calibration loss with high probability.

Fourth, it did not feel like very much attention was paid to the probabilistic nature of the requirement.
In related fields like differential privacy, say, a number of epsilon/delta guarantees are available, but with specific values of epsilon and delta that are prohibitive in real data contexts.
I would like to see an empirical examination of the bounds in your theorems under realistic contexts.

Fifth, the paper is currently unclear about the training/calibration/test data split.
For UCI you say 25% is used for testing and the rest for training.
How is the training data split between the training data and calibration data?
Or is the test data being used to calibrate?
If so, I'm afraid that would be problematic.
Similarly, please be explicit about how this is done for the image data.

Sixth, for UCI I would like to see a more thorough assessment of the relative strengths of the different methods.
You write that your method improves on prior work in many places "without ever being worse", but that doesn't obviously seem to be the case (e.g., Deep ensemble mpg where your method seems worse than an uncalibrated baseline).
It's quite hard to make sense of giant tables of numbers like this, so it might be helpful to do some plots or post-processing to tease out the relationships.

Seventh, and a minor point that doesn't affect my score, there are a number of typos (e.g., "computing *extract* predictive uncertainties", "distirbutional") so the paper could probably use another pass. The phrase "w.h.p." is also introduced without being defined, but is actually important to discuss (I assume you mean 'with high probability').

**Summary Of The Paper:**

The authors propose an algorithm scheme that they prove provides probabilisitic asymptotic guarantees of `distribution' calibration for models.
The algorithm provided by the authors fits a primary model, H, on a training dataset, D, as well as a recalibrator, R, which takes H(x) as an input, using a calibration dataset, C.
H is fit using (x,y) pairs from D and a proper loss. R is then fit using (H(x), y) pairs from C and a proper loss.
The resulting model ought to be calibrated on C while also being (asymptotically) as sharp as the original model.
The authors also introduce the idea of calibrated risk minimization, which is a constrained optimization alternative to empirical risk minimization.

**Summary Of The Review:**

I do not currently recommend acceptance.
My main concern is some conceptual clarity about what different data-sets are doing which is reflected both in the theoretical and experimental sections.
I would like to see more thorough examination of the probabilistic bounds and the degree of slack they afford, which could help better ground the claims that calibration is a free lunch and justify CRM as an approach.

---

### Official Review · Reviewer_fwct · 2021-11-01

**Correctness:** 3
**Technical Novelty And Significance:** 2
**Empirical Novelty And Significance:** 2
**Recommendation:** 3
**Confidence:** 4

**Main Review:**

Strengths:

- The paper proposes a simple recalibration algorithm
- The proposed method works if the number of outputs is reasonably small (one or two)
- Extensive simulation

Weaknesses:

- Unclear / small novelty compared to previous work
- Theoretical results are asymptotic
- Empirical results are quite weak

I have previously served as a reviewer for this paper for NeurIPS 2021. We ended up rejecting the paper there and recommended a few major points for improvement:
- Performance in higher dimensions: The current method seems to only work well in very low-dimensional settings (one to two dimensions or so). It should be discussed how it would (or would not) generalize to higher-dimensional problems, which are relevant and common in many areas.
- Empirical results: The current results are missing error bars, which makes it impossible to assess their statistical significance. In fact, the proposed method underperforms the baselines in many settings, which may or may not be significant.
- Missing baselines: Previous work has proposed to optimize specialized calibration losses, such as [1,2,3]. It would be necessary to compare to at least some of these related approaches to judge whether the proposed approach actually improves over prior work.

Given that it seems like the authors have unfortunately implemented none of these suggestions yet, I will for now have to stick to my previous assessment of the work and recommend rejection again.



[1] Khosravi et al. 2011, https://ieeexplore.ieee.org/document/5672788

[2] Pearce et al. 2018, https://arxiv.org/abs/1802.07167

[3] Tagasovska et al. 2018, https://arxiv.org/abs/1811.00908

**Summary Of The Paper:**

The authors propose to decompose different common loss functions into calibration and refinement terms and show theoretically that under some (strong) assumptions, such as universal approximation, models can become perfectly calibrated without sacrificing predictive performance. They propose a constrained optimization framework to achieve such results and demonstrate it empirically on different tasks.

**Summary Of The Review:**

This work has previously been rejected from NeurIPS 2021 due to the absence of a discussion on the performance in higher dimensions, the lack of errors bars in the experimental results, and missing baselines. Since none of these points have improved in the resubmission, I will again recommend rejection.

---

### Official Review · Reviewer_qXzz · 2021-11-03

**Correctness:** 2
**Technical Novelty And Significance:** 2
**Empirical Novelty And Significance:** 3
**Recommendation:** 5
**Confidence:** 3

**Main Review:**

The paper addresses an important problem, namely that of ensuring that uncertainties output by supervised machine learning models are well calibrated. It is largely well written and well motivated. The method is simple and generally applicable. It mainly represents a generalization of prior recalibration approaches. Empirical results on the UCI regression datasets show that the model consistently improves calibration compared to the base model, and modestly improves over previous work by Song et al. It also shows strong improvements on image classification tasks.

My main concern with the paper is that its theoretical analysis appears somewhat tautological to me. It starts with the assumption that the calibrator R is close to Bayes-optimal, and shows that in this case, it ensures calibration without hurting sharpness. But given that R was trained on the same loss, and on the same data distribution as the base model H, why should it be any closer to Bayes optimal than H? Since the analysis makes no assumptions regarding the nature of R or H, it seems difficult to make any definitive statements in this regard. I suspect the answer is that, in practice, R will have much fewer parameters than H and be less prone to overfitting. It would be important for the paper to analyze the specific conditions under which recalibration is effective, instead of simply asserting that it always works.

Specific Questions/Comments:
 - I am confused by the definition of the check score in section 2.2. It appears that f denotes an outcome from Y, as opposed to a probability distribution function over y, which is how it was used previously.
 - The theoretical analysis states that base model and calibrator are trained on different training sets, but the experimental section does not mention anything in this regard. Were the sets actually diffirent, and if so, what was their size? Do you think this is an important setting?
 - It is unclear to me to what degree the experimental outcomes are effected by the increased capacity of the recalibrated model. It would be good to provide a baseline in which R and H are jointly trained on all the training data.
 - There are a few broken sentences, e.g. at the end of the model-based planning paragraph in section 5.2.

**Summary Of The Paper:**

The paper proposes a new method for calibrating the uncertainty estimates of supervised machine learning models, i.e., ensuring that they match the probabilities of the data distribution. After regular model training, a calibrator model is trained to transform the model output, using the same loss but a separate dataset. Theoretical analysis and empirical experiments are conducted to support the claim that this is a way to calibrate "any model" at "almost no cost".

**Summary Of The Review:**

In summary, the paper provides a simple method that empirically generally improves model calibration. However, the theoretical analysis is overly general and does little to identify the specific conditions under which is works. As a result, I am skeptical of the very broad claim that it will be effective on any model and learning problem.

---

### Official Review · Reviewer_EFrC · 2021-11-04

**Correctness:** 1
**Technical Novelty And Significance:** 2
**Empirical Novelty And Significance:** 2
**Recommendation:** 3
**Confidence:** 2

**Main Review:**

**Significance and contribution**


In the abstract, introduction and theoretical analysis there are some ambitious claims such as “This paper introduces algorithms that ensure the calibration of any model while maintaining sharpness”,  “our algorithms achieve for the first time a long-standing statistical principle that forecasts should maximize sharpness subject to being calibrated”, and “in a sense, calibration can be a free lunch“. In my view, these are unsubstantiated.

Methodologically speaking, it is not clear to me what is the contribution of this paper. The paper states that it proposes a framework for recalibration, within which any flexible model class can be used as a recalibrator.  In practise, as far as I can tell, one important difference from previous works is the use of NNs with a simultaneous quantile regression noise model as the recalibrator.

The theory section contains a series of propositions regarding asymptotically optimal calibration. My impression is that these are somewhat vacuous: we would not usually have an infinite amount of held-out data to train an infinitely flexible recalibration model. Again, I think that saying that “calibration can be a free lunch“ based on this is a bit wishful. It is like saying that machine learning is a free lunch because we have asymptotically optimal learners.



**Clarity and Writing**

The introduction and background sections of the paper read very well. The problem is explained and motivated well. I would say that simple concepts are explained in an unnecessarily notation-heavy way at times.

I was left somewhat confused by the methodological section (3). It reads a bit like a literature review. I did not understand what the methodological contribution of the paper was from reading this section.

I was also somewhat confused by section 5. Here “Calibrated Risk Minimization” is introduced in definition 5.1 as a principle by which the predictive entropy is minimised while miscalibration is kept at a minimum. As far as I can tell, no procedure is actually given to obtain this result.
Section 5.2 also feels disconnected. It gives a series of example settings in which calibrated uncertainty would be useful. I would suggest moving this information to the introduction or related work.

**Experiments**

I think key details are missing from the experimental section. It is unclear to me what the exact procedure used to recalibrate predictive distributions is. Did the authors employ the standard procedure of training a secondary model on some held out test data using a standard objective? Or did they employ some more sophisticated method designed to adhere to the “Calibrated Risk Minimization” principle introduced in 5.1?

I could not find what percentage of training data was used for recalibration in the prose. I think the authors should mention this, as it is an important hyperparameter.

I would recommend bolding important figures in the tables. In its current state, table 2 looks like a matrix of 288 numbers. It is very difficult to read. I would also encourage the authors to repeat their experiments with different random seeds and report error bars.

**Summary Of The Paper:**

This paper proposes to re-calibrate the probabilistic predictions of machine learning models by learning a secondary function that maps from their output parameters to a new set of calibrated distribution parameters. The secondary function is learnt on a held out set. The paper argues for the use of flexible recalibration methods, appealing to their asymptotic optimality. Experiments are performed on 8 UCI regression datasets, MNIST, SVHN and CIFAR10.  NNs are used as both primary and recalibration models. Superior results are obtained compared to some recent baselines.

**Summary Of The Review:**

Although I do not have a large amount of research experience in the domain of calibration I do have quite a bit of applied experience in this domain. For that reason, after reading the introduction and abstract I was quite excited about this paper as I was expecting some sort of theoretical breakthrough. After reading the rest of the paper I was just left confused. Unless I am misunderstanding something important, I think there is a large mismatch between the claims this paper makes and the theoretical and empirical results it delivers.

---

### Decision · Program_Chairs · 2022-01-20

**Decision:**

Reject

**Comment:**

The reviewers highlight that several of significant claims of the paper are not backed up by experiments, and the experiments themselves lack sufficient detail, therefore, at this stage, I recommend rejection. I suggest the authors address the questions and comments they have received before considering whether they might resubmit or not.